# Effect of Substrate Negative Bias on the Microstructural, Optical, Mechanical, and Laser Damage Resistance Properties of HfO_2_ Thin Films Grown by DC Reactive Magnetron Sputtering

**DOI:** 10.3390/mi14091800

**Published:** 2023-09-21

**Authors:** Yingxue Xi, Xinghui Qin, Wantong Li, Xi Luo, Jin Zhang, Weiguo Liu, Pengfei Yang

**Affiliations:** School of Optoelectronic Engineering, Xi’an Technological University, Xi’an 710021, China; 18792801300@163.com (X.Q.); 18392382339@163.com (W.L.); luoxi202307@163.com (X.L.); j.zhang@xatu.edu.cn (J.Z.); wgLiu@163.com (W.L.); pfyang@xatu.edu.cn (P.Y.)

**Keywords:** HfO_2_, reactive magnetron sputtering, negative bias, optical properties, laser-induced damage threshold

## Abstract

Hafnium oxide thin films have attracted great attention as promising materials for applications in the field of optical thin films and microelectronic devices. In this paper, hafnium oxide thin films were prepared via DC magnetron sputtering deposition on a quartz substrate. The influence of various negative biases on the structure, morphology, and mechanical and optical properties of the obtained films were also evaluated. XRD results indicated that ($\bar{1}11$)-oriented thin films with a monoclinic phase could be obtained under the non-bias applied conditions. Increasing the negative bias could refine the grain size and inhibit the grain preferred orientation of the thin films. Moreover, the surface quality and mechanical and optical properties of the films could be improved significantly along with the increase in the negative bias and then deteriorated as the negative bias voltage arrived at −50 V. It is evident that the negative bias is an effective modulation means to modify the microstructural, mechanical, and optical properties of the films.

## 1. Introduction

Hafnium oxide (HfO_2_), frequently employed in high-refractive-index material with wide transparency from ultraviolet (UV) to midinfrared (MIR), is an important optical thin film material due to its excellent laser damage resistance, high hardness and outstanding thermal and chemical stability [1,2]. In recent years, hafnium oxide has received more attention in semiconductors as a high-k dielectric film grown on silicon substrates. Hafnium oxide has many advantages, such as a wide band gap (E_g_ > 5.0 eV), a relatively high dielectric permittivity (~25) and good thermodynamic stability, which makes it the most promising material to replace silicon-based gate oxides [3,4]. Moreover, hafnium oxide with an orthogonal phase is also a remarkable ferroelectric film material and has potential applications in the field of non-volatile memory fabrication [4,5]. It has been proven that the formation of the orthogonal phase causes ferroelectricity and the phase structure be regulated by the component and procedure [6,7,8,9].

Generally, hafnium oxide films can be prefabricated by utilizing chemical vapor deposition [10], electron beam (EB) evaporation [11], magnetron sputtering [12,13,14], atomic layer deposition [15,16], and pulsed laser deposition [17]. Among them, the atomic layer deposition (ALD) technique has become the major preparation method for hafnium oxide because of its low temperature, self-limiting surface reaction mechanism, and atomic-level thickness. Unfortunately, some impurities such as carbon and nitrogen can be observed in the films deposited using the ALD technique [18]. Comparatively, uniform and dense films with stable components and high purity can be obtained and the metal electrodes prepared synchronously by using the magnetron sputtering technique. The quality of films can be optimized further by adjusting the working pressure, target power, gas ratio, substrate temperature, and negative bias [19].

Negative bias applied to the substrate has often been utilized in various magnetron sputtering deposition devices [19,20,21] so that the flux and energy of depositing charged species can be varied. The effect of negative bias upon the energy of ionic species appearing in the magnetron plasma is also very appreciable, which consequently affects the total energy delivered to the substrate during deposition [22]. Via this approach, the growth process can be modified, therefore probably determining the optical and microstructural properties. Nezar S. et al. [23] demonstrated experimentally that the deposited titanium dioxide thin films shift from a single anatase phase to a polycrystalline structure with a mixture of anatase and rutile phases when applying a negative voltage on the substrate with a range of 0 to −100 V. For HfO_2_ thin films deposited by RF magnetron sputtering with pulse DC substrate bias, Maidul Haque et al. [24] reported that the density of the films deposited with substrate bias is detected to be higher than the density of the films deposited without substrate bias due to Ar^+^ ion bombardment. Nahar et al. [25] conducted research on substrate bias’s effect on HfO_2_ thin films and found that the electrical properties of HfO_2_ were enhanced and lower leakage current and higher breakdown voltage were exhibited after applying substrate bias. Nevertheless, the effect of negative bias in the case of deposition of HfO_2_ films by DC (direct current) magnetron sputtering has not been reported hitherto.

Notwithstanding that some of the details need to be further understood and studied, there exists little doubt that substrate bias controls the gas and void contents of the thin film [26]. Therefore, it is found that the incorporated gas atoms are proportional to the square of the voltage value of the substrate negative bias [27]. Additionally, the adsorbed gas of the growing film surface may be re-sputtered, resulting in variation in the density of the thin film during low-energy ion bombardment. HfO_2_ or HfO_2_-based thin films require electrical properties that can be produced with fewer defects and voids via the application of a negative bias voltage to the substrate [19,25]. This paper presents a series of experiments where HfO_2_ thin films were prefabricated via the DC magnetron sputtering technique at different mains of substrate negative bias voltages. In the meantime, the effect of substrate negative bias voltage on the structural, optical, and mechanical properties and laser-induced damage thresholds of the HfO_2_ films is systematically explored. The final results are also presented in the paper in the last section.

## 2. Experimental Procedure

To begin with, using the DC reactive magnetron sputtering method, HfO_2_ thin films were deposited on quartz substrate without substrate negative bias but with different substrate negative biases. A high-purity hafnium target (99.9%) of 60 mm in diameter and with a thickness of 5 mm was placed beforehand as a sputter target. Prior to deposition, the vacuum was raised to 2 × 10^−3^ Pa. Next, 40 sccm of high-purity argon (99.99%) gas was introduced and pre-sputtered for 15 min to remove oxides and other impurities from the target surface. Thin Hf targets were sputtered into pure oxygen (99.99%), without argon as a working gas. The working pressure in the deposition chamber was set to 0.6 Pa. The substrate was placed on a tungsten-heated substrate table and the substrate temperature was set at 200 °C by a thermocouple. During the deposition process, the distance between the target and substrate was set at 50 mm and the power density of the target was set at 5.3 W/cm^2^ in advance for each hafnium film. The thickness of the thin films was configured using a deposition time equivalent to 40 min for all samples deposited at different substrate negative biases. The thicknesses of the samples deposited were measured using a surface profilometer from ZYGO, which was around 160 nm. Additionally, the voltage applied to the substrate was set to be varied (0, −25, −50, −75, and −100 V). However, simultaneously, all other deposition parameters were maintained in constancy so as to investigate the effect of negative bias on the crystalline structure as well as the properties of the films.

X-ray diffraction (XRD-7000 type, from Shimadzu, Japan) was utilized to explore the structure of the deposited films. GIXRD measurements were carried out via Cu Ka X-ray radiation (1.54 Å), along with 2*θ* detector scan with the incident beam preset fixed at a constant grazing angle of 1°, a scanning rate of 5°/min, as well as a scanning range from 20° to 85°. The film surface’s morphology was measured via an atomic force microscope (AFM, from Bruker, Mannheim, Germany) in peak-force tapping. The films’ optical transmittance was measured via a UV–vis–NIR spectrophotometer (U-3501type from Hitachi) in a wavelength range from 200 to 1100 nm. The band gap of the film was obtained from the transmission spectrum. The refractive index was determined via a J.A. Woollam M-2000UI spectroscopic ellipsometer (SE). All data measured via the process mentioned above were collected to design an optical model that is of use for obtaining the thickness and optical properties via performing regression analysis. The optical model consists of four phases (from bottom to top), which are the silica substrate, the bulk HfO_2_ film, the surface rough layer [28] composed of 50% void space as well as 50% HfO_2_, and the incident medium (air) utilized to explore the HfO_2_ thin films. The four-layer model used in the Wvase32 software (Version 3.335) is shown in Figure 1. In this paper’s analysis of SE data, the unknown dielectric function of HfO_2_, which has only a small amount of absorption in the visible and near-infrared regions, is described by the Cauchy–Urbach model, as shown in the following two formulas:(1)$$n\left(\lambda \right)={\;A}_{n}+B_{n}/\lambda^{2}+C_{n}/\lambda^{4}$$
(2)$$k\left(\lambda \right)=\alpha e^{\beta (1240/\lambda -E_{g})}$$
where *A_n_*, *B_n_*, and *C_n_* are index parameters that specify the index of refraction and *α*, *β*, and *Eg* are the extinction coefficient amplitude, the exponent factor, and the band edge *g*, respectively, and are defined as variable fit parameters during the process of evaluating the data [29]. The incidence angle was set at 75° during the S.E measurement.

The nano-hardness and elastic modulus of the film were measured by the continuous stiffness method (CSM) using a nanoindenter (Agilent G200, Agilent, Milpitas, CA, USA) fitted with a Berkovich indenter diamond tip with a radius of curvature of 20 nm via a continuous stiffness measurement mode (CSM). The obtained hardness value for each sample was the average value calculated after 10 measurements in several randomly selected regions on the film surface. The LIDT measurement of the films was carried out in a “1-on-1” mode manner [30] according to the international standard ISO11254-2 [31] and employing a 1064 nm Q-switch pulsed laser at a pulse length of 12 ns and beam size of 80 mm. The experimental characteristics and its process for determining LIDT from damage probability plots have been comprehensively explored in the earlier works [32].

## 3. Results and Discussion

Owing to the function of the negative bias, some changes could be observed in the crystalline structure of hafnium oxide thin films, which were examined by XRD measurement. Figure 2 shows the results. The main diffraction peak was observed at approximately 28.3° with or without the application of the negative bias, suggesting that the crystallization in the films is caused by diffraction from ($\bar{1}11$) planes of monoclinic phases of HfO_2_ [33]. Moreover, the other less dominant diffraction peaks are ascribed to the orientations (200), (020), (002), and (022). In addition, (221) appeared in the diffraction patterns. No diffraction peaks from the tetragonal or cubic phase were obtained. It was revealed by the XRD patterns that as the negative bias increases from 0 to −100 V the intensity of the dominant peak ($\bar{1}11$) decreases, which is an indication of a decline in the average crystallite size of the films.

Although the intensity of other less dominant peaks increases to different degrees along with the increasing of negative bias, a clear ($\bar{2}11$) diffraction peak was obtained in samples deposited at a negative bias voltage of −100 V. The preferred orientation of thin films is associated with the competitive growth mechanism [34]. The energy of bombarding ions can be appreciably increased by a negative bias voltage placed on the substrate, which provides more energy to competitive growth among planes at different orientations. The Debye–Scherrer formula is introduced here to estimate the average crystallite size [35] as follows:(3)$$D_{hkl}=\frac{k\lambda }{\beta cos\theta }$$
where *D_hkl_* is the grain size, *k* = 0.89, the X-ray wavelength *λ* is 1.54 nm, and *β* is the diffraction peak’s half-height width and the diffraction angle. Based on the value of the line width (FWHM) of the most intense peak, the average crystallite size was estimated. There existed a connection between the line width (FWHM) and ($\bar{1}11$) of the monoclinic phase, and the line width (FWHM) was obtained via fitting the diffraction peak to the Lorentzian distribution. As the substrate bias voltage varied from 0 to −25 V or −50 V, the grain size decreased slightly from approximately 84nm to 81nm, which suggested a grain refinement. As the substrate bias voltage was increased to −75 V, the grain size continued to decrease to 79 nm and then dropped to 76 nm as the negative bias voltage increased to −100 V. It can be seen that the grain size tends to decrease as the negative bias voltage increases in the range 0 to −100 V.

The two-dimensional AFM images of the films deposited under biased or unbiased conditions by the reactive rf magnetron sputtering technique are shown in Figure 3a–e. The surface morphology of the deposited films was markedly affected by the substrate bias voltage. From this figure, it can be obviously observed that the film deposited on unbiased substrate during sputtering shows higher surface roughness with larger grain size compared with the films deposited with low negative biases. The film deposited at −25 V substrate bias voltages is observed to have a smooth surface and lower surface roughness, which was homogenous and coincided with an average grain size of 81 nm. When the substrate voltage increased to −50 V, the grain size was decreased to the same grain size according to XRD data in Figure 2. The surface morphology of the films deposited at −50 V substrate bias voltages was found to be almost identical despite having the occurrence of slightly more voids. When the substrate bias voltage was increased to −75 V, this film consisted of fine grains, which was likely ascribed to the formation of more grain boundaries and voids caused by energetic-particle bombardment during deposition. It has been previously established by pertinent research that when bias is applied to the substrate, the substrate attracts energetic ions and neutrals. The bombarding particles re-sputter loosely bonded atoms from the deposited film [25]. This process reduces the intrinsic defects and thus increases the film density. However, for these films deposited at high or overall oxygen partial pressure, the void fraction in the films shows an overall increasing trend, impeding the density from increasing as the value of negative substrate bias increases further [24]. This is due to the result of negative oxygen ions, where low-energy negative oxygen ions are unable to reach (impact) the substrate table sample in the presence of a negatively biased electric field in the substrate, leading to an increase in the oxygen void content in the films. The grain morphological evolution of all the HfO_2_ films can be explained via the variation of low-energy ions and neutrals by the application of different substrate biases. As the substrate bias was changed from −25 V to −100 V, the root mean square roughness (Rrms) increased from 2.89 nm to 4.63 nm. This result may again be related to the variation in the low-energy distribution of ions and neutrals bombarding the grown film modified by the substrate bias voltage.

The transmission spectra were measured as a function of the wavelength of the incident light so as to investigate the effect of substrate negative bias on the optical properties of HfO_2_ thin films. The transmittance spectra of all the samples deposited at different negative biases, as well as quartz substrates, were registered as a function of wavelength, which is shown in Figure 4. The oscillation of the spectrum with wavelength is attributed to interferometric (constructive and destructive) effects [36]. Each spectrum reveals the number of peaks and valleys at a fixed wavelength (*λ*) concerning the optical thickness, which is the result of the product of the refractive index of the deposited film and the physical thickness. The change in refractive index is almost constant for the same material, so the optical thickness of the film is proportional to the physical thickness. As a result, the qualitative thickness of the deposited film could be estimated according to the measured transmission spectra. The lower transmittance of films deposited without negative bias might be attributed to defects and absorptions in HfO_2_ film, which effectuates an increase in the absorption of visible light, then brings about a decrease in transmission. The films deposited under −25 V and −50 V show higher transmittance. The improvement in the transmittance of films deposited at different lower negative biases indicates that the defects and absorptions of films have been suppressed owing to low-energy ion bombardment. Although the details need to be further explored and studied, the growing films have been effective in modifying surface roughness, grain size, and defects during a broad low-energy distribution of ion and neutral bombardment. However, it can be noticed that the transmission was decreasing as the negative bias was increased from −50 to −100 V; the level of degree of reduction in transmission is the largest at a bias voltage of 100 V. This result may be related to grain refinement and the increase in film defects as the bias voltage on the substrate increases. Additionally, as the substrate bias was changed from −25 V to −100 V, the root mean square roughness (Rrms) increased significantly, which introduced the sources of light scattering. The physical thickness of the deposited films was qualitatively estimated by transmission spectrum using the envelope method [37]. The thickness of the film was measured by a surface profilometer, and the thickness of each film was measured/calculated by different methods approximated to each other, regardless of the method used. By analyzing the transmittance spectra, the change in the negative bias value caused only a small drift in the transmittance spectra, which might be related to the change in the refractive index of the film, indicating that the applied negative bias did not significantly vary the thickness and deposition rate of the deposited films.

The optical band gap of the deposited films at different negative biases can be calculated using the Tauc method [38]. Figure 5 shows the absorption spectra and Tauc’s plot of the HfO_2_ thin films deposited at different negative biases. The energy band gap (*Eg*) value of the HfO_2_ was obtained by absorption spectra and plotting (*αhν*)^1/2^ vs. photon energy (*hν*) via the following equation:$$\alpha \left(h\nu \right)={A\left(h\nu -E_{g}\right)}^{n}$$
where *A* is a constant, α is the optical absorption constant, *hν* is the incident photon energy, h is Planck’s constant, and the index *n* characterizes the electron leap characteristics and is taken according to the band gap type. For a direct band gap, *n* = 1/2, in this case, the curve (*αhν*)^2^ is linearly related to the photon energy (*hν*) in a certain photon energy range and the line made with the *X*-axis intercept can give the exact value of the optical band gap; for an indirect band gap *n* = 2, the band gap can be approximated by a linear fit of the curve (*αhν*)^2^ to the photon energy (*hν*) with the *X*-axis intercept. It is widely acknowledged in the literature [39] that HfO_2_ is an indirect band gap dielectric material with *n* = 2. The linear part of the derived curve extends to the hν axis with an intercept of (*αhν*) = 0, which gives rise to an approximation for *Eg*.

From the inset in the upper left of the figure, it can be clearly seen that the optical bandgap ranges from 5.75 nm to 5.79 nm when the substrate bias is changed from 0 V to −100 V. As previously reported, the bandgap energy of monoclinic-phase HfO_2_ films is 5.41 eV~5.86 eV, which is close to our results. It is known that the optical band gap is affected by numerous factors, such as defect density, purity, packing density, stoichiometric ratio, and grain size. According to Figure 5, the blue shift in the optical band gap has been observed along with the increase in the negative biases from 0 V to −50 V. The decrease in crystallite size is a key factor in causing the increase in optical band gap energy along with the increase in the negative biases. Some researchers hold the opinion that the quantum size effect weighs heavily in results and an increase can appear in the band gap energy, especially if the crystallite size is less than 30 nm. Although the average crystallite size of HfO_2_ deposited in this study is much larger than 30 nm, in accordance with the results of XRD and so on, it seems that the quantum confinement effect can slightly modify the band gap energy. The crystallite size decreases slightly as the negative biases increase from 0 V to −50 V, hence the decreasing crystallite size increases the band gap energy of thin films. However, as the substrate voltage increased to −50 V and −100 V, the band gap energy for the deposition was decreased to about 5.76 and 5.75, respectively. This phenomenon can be explained by the following reason: the defects and voids in HfO_2_ thin films increase with the increase in the negative bias voltage from −50 V to −75 and −100 V, and this is due to the decrease in the transmittance in the UV–vis region of the films, as supported by transmittance spectra data. Since the variation in the bandgap with negative bias is only within the range of 0.04 eV, the results in this paper are available for reference, and the effect of negative bias on the bandgap needs to be analyzed more precisely.

In order to substantially investigate and describe the optical constants such as refractive index (n) and extinction coefficient (k) of the HfO_2_ films deposited under various substrate voltages we conducted further experiments. It was by fitting the experimental data (psi (ψ) and delta (Δ)) obtained from spectroscopic ellipsometry that the optical constants were eventually determined using a four-layer model (air, roughness, film, and substrate) in Wvase32 software. The obtained refractive indices and extinction coefficient plots as a function of wavelength in the VIS/NIR region are depicted in Figure 6a,b. The obtained results for both n and k apparently show an exponential and dramatic decrease with an increase in wavelength, even though it is maintained almost in constancy at higher wavelengths for all the deposited films. The refractive indices for the as-deposited thin films under 0 V, −25 V, −75 V, and −100 V substrate biases are respectively found as 2.01, 2.03, 2.03, 1.99, and 1.97 for λ = 550 nm. These results are greatly in accordance with the values reported earlier.

It is known that packing density is the decisive factor affecting the refractive index. The packing density, *p*, of a porous film is defined by Yelda’s [40] formula as follows:$$p=\frac{\left(n_{film}^{2}-1\right)}{\left(n_{bulk}^{2}-1\right)}$$

The value of the refractive index *n_bulk_* of hafnium oxide in the bulk is about 2.1. The value of *n_film_* is the refractive index of thin films, which is around 600 nm for all samples [29]. The values of packing density for films deposited under different biases have been calculated as: 0.88, 0.90, 0.91, 0.86, and 0.83. The packing density is linked to the thickness of thin films, crystalline structure, and crystallite size. According to the results of the transmittance spectra in Figure 4, the negative bias has a rather minor effect on the film thickness, whereas the XRD results in the Figure 2 also confirm the grain refinement along with the increasing negative bias voltage.

As seen from the figure, compared with the deposition of hafnium oxide films on unbiased substrates, the refractive index of the deposited films increases significantly when a negative bias is applied at −25 V or −50 V, which indicates that the grains and defects of the growing films decrease weakly while the increment in the refractive index was detected along with the increase in substrate biases, which can be attributed to the increased packing density owing to low-energy ion and neutral bombardment. These optical constant results can be correlated with the microstructure and packing density of the films. The films with low packing density have a low refractive index, whereas compact and dense microstructures with smooth surfaces retain a high refractive index. It is observed that a further increase in the negative bias induced the reduction in the refractive index owing to a low value of packing density at a high negative bias voltage.

All samples’ extinction coefficients (k) approximate to zero and keep very low in the visible region. It is perceived that the trend in the extinction coefficients of samples along with the increasing of applied negative bias is opposite to the trend for the refractive index. The extinction coefficient encompasses the contribution from the absorption and the scattering of grains. Higher negative biases result in decreased packing density and the size of the grains in thin films (as shown in Figure 3), which enhances the scattering effect in HfO_2_ thin film.

Figure 7 shows the test results of the mechanical properties of the deposited films at different negative substrate biases using the nanoindentation technique. The nano-hardness and elastic modulus of the deposited films was appreciably affected by the substrate bias voltage. The data in Figure 7 reveal that a slight increase appears in the obtained average values for hardness and modulus from 8.6 to 8.9 GPa and another from 136.4 to 140.5 GPa, with an increase in negative substrate bias from 0 to −50 V. These results have brought about the conclusion that the increasing bias voltage was conducive to the hardness of deposited films under lower substrate bias conditions and can be attributed to its density. The introduction of voids caused an increase in the porosity and a decrease in the packing density of thin films, which resulted in the reduction of nano-hardness and elastic modulus, as reported in the literature. As mentioned earlier, when the negative bias voltage is varied from 0 to −50 V, the refractive index of the film increases while the void fraction decreases. This is due to the fact that enhanced bombardment of Ar^+^ ions on the growing films in the negative bias leads to more compact and denser films [24]. However, the thin film deposited under −75 V substrate bias revealed a hardness of 8.3 GPa and an elastic modulus of 130.21 GPa. Further increase of the substrate bias to −100 V resulted in a decrease of hardness to 7.9GPa and elastic modulus to 126.6 GPa. Such results are likely to bring about the conclusion that the increase of substrate bias was not favorable for the hardness of deposited films under higher substrate bias conditions. As indicated by the refractive index and the void fraction of HfO_2_ films obtained from the transmission measurements and ellipsometry analysis, the refractive index of the films shows an overall decreasing trend.In the same tine he void content in the films shows an opposite trend with an increase in substrate bias in the range of −50 V to −100 V owing to a reduction in density. In short, the less dense the thin film is, the less high hardness could be obtained.

The elastic modulus of the films is mainly affected by the internal stress and microstructure of the films. Considering the limitation of the space of this paper and the experimental conditions, the variation of the film’s internal stress is not particularly discussed. However, when comparing the elastic modulus to the packing density of HfO_2_ thin films deposited under different negative biases. it can be seen that both of them reflect the similar trend along with the increase of negative bias pressure, i.e., the greater the packing density, the higher the elastic modulus of the film.

Figure 8 shows the laser-induced damage threshold (LIDT) fitting curves of HfO_2_ thin film deposited at different substrate negative biases. From the Figure 8, it can be observed that the LIDT value increases along with the increase of substrate bias voltage in the range of 0~−50 V, and decreases along with the increase of bias voltage when the substrate bias voltage is further increased to −75 Vand −100 V. The LIDT value for unbiased films is 13.19 J/cm^2^ and it increased to 13.46J/cm^2^ and 13.42 J/cm^2^ along with the increasing of substrate bias voltage to −25 V and −50 V, respectively. Jena et al. [30] have reported that the LIDT value varies from 9.89 to 8.83 J/cm^2^ for HfO_2_ thin films deposited by electron beam evaporation technique at different oxygen (O_2_) partial pressures, and this result is tested by a 1064 nm pulsed laser (7 ns pulse width), which is comparatively less than our results.

Thermo-mechanical damage process is regarded as the main damage process of dielectric thin films under nanosecond pulsed laser irradiation [41]. It is commonly believed that defects such as grain boundaries, porous sites, nonstoichiometric defects, inclusions as well as nodules [42] in dielectric thin films absorb the pulsed laser irradiation, and the excess thermal energy is coupled with the optical structure, which results n damage. The thermophysical properties of the dielectric thin film, such as density, specific heat, and thermal conductivity, all play key roles in limiting LIDT. According to the mechanism of defect-originated thermal damage, the LIDT may be fabricated as the following formula, [43]
$$F_{th}\approx \frac{16T_{C}}{\pi }{\left(\rho_{film}C_{film}K_{film}\tau \right)}^{1/2}$$
where *F_th_*, *T*_C,_ *ρ_film_*, C_film_, *K_film_* are the damage threshold in J/cm^2,^ melting temperature, density, specific heat at constant pressure and thermal conductivity of the film respectively and *τ* is the laser pulse length.

As a matter of fact, for dielectric oxide films, the packing density plays the most important role in improving their resistance to laser damage. The thermal conductivity and specific heat of the film increase along with the increase of film density as proved by S. Jena [30]. Denser films with lower thermal barriers take on superior thermal conductivity and hence is provided with a higher laser damage threshold. Compared with the refractive index and laser damage threshold plots of the deposited films under different substrate negative biases, a phenomenon can be found that their values have a similar variation trend along with the increasing of negative bias. And the reason may be the packing density. Denser packing density may effectuate a compact packing structure which conversely increases the refractive indexes of the film with the same number of particles condensing on the substrate. However, the LIDT values of HfO_2_ films decreased slightly as substrate negative biases varied from −25 V to −50 V, and the reason may be the surface roughness. The nanoscale or sub-nanoscale defect, which is correlated to the surface roughness, is inclined to evolving into microscale damage at the film surface [44], thereby, the LIDT values decrease with the surface roughness of thin film.

## 4. Conclusions

In summary, this paper reports the effect of substrate negative bias on the structure, surface morphology, optical properties, mechanical properties, and laser-induced damage threshold of the DC magnetron sputtering deposited HfO_2_ thin films. The films were eventually discovered to be polycrystalline predominantly composed of monoclinic crystal structure and the grain sizes degraded with an increase in negative bias during deposition. AFM images show that the HfO_2_ films are of high quality, with a dense uniform grainy morphology. The slight modification in grain refinement and packing density are found to exist in the HfO_2_ films along with the increasing of negative biases, which is owing to the bombardment of the growing films by ions and neutrals with a broad low-energy distribution. The experiments have unfolded that substrate negative bias affects the diverse properties of the films. Compared with HfO_2_ thin films deposited without substrate negative biases, more compact and denser films were of −25 and −50 V voltages. However, as the negative bias is increased to −75 V or −100 V, the density decreases inversely along with the increase of the negative bias voltage. The peak transmittance, refractive index, band gap, nano hardness, elastic modulus, and LIDT of HfO_2_ films show similar trends along with the increase in negative bias. Therefore, the variation of density and void content in the films is eventually reflected in the variation of overall properties of the HfO_2_ films deposited at different negative biases.

## Figures and Tables

**Figure 1 micromachines-14-01800-f001:**
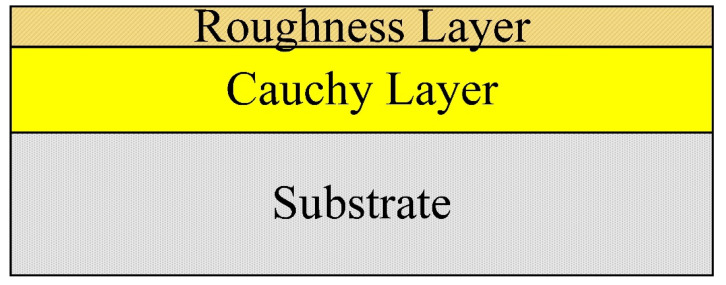
Ellipsometry-fitted physical model schematic diagram (the unknown dielectric function of HfO_2_ is described by the Cauchy layer and roughness layer).

**Figure 2 micromachines-14-01800-f002:**
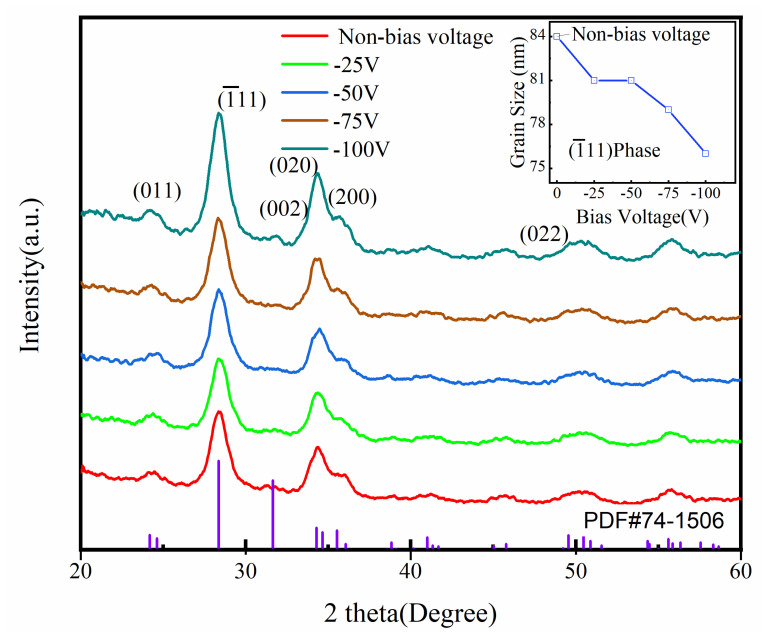
X-ray diffraction patterns of the HfO_2_ films deposited by DC reactive magnetron sputtering at different negative bias voltages. The crystallite sizes have been calculated by using the ($\bar{1}11$) crystal plane of HfO_2_ films deposited at different negative bias voltages inserted in into the upper right corner of this graph.

**Figure 3 micromachines-14-01800-f003:**
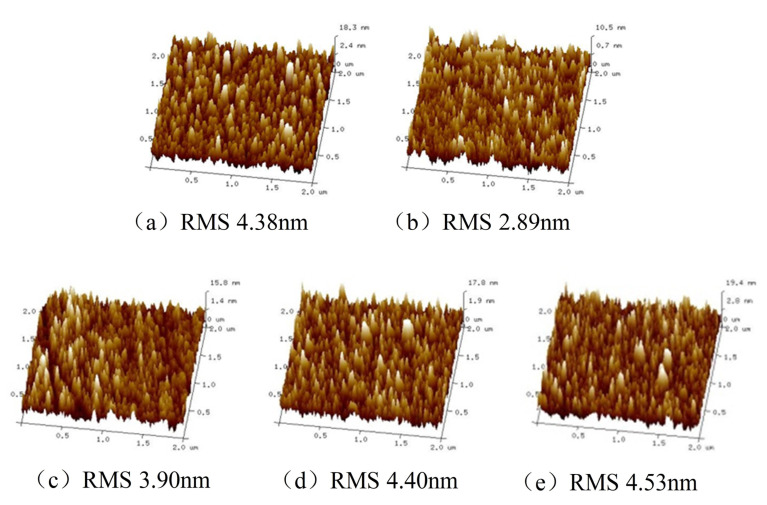
AFM micrographs of hafnium oxide films deposited at (**a**) non-bias voltage; (**b**) −25 V; (**c**) −50 V; (**d**) −75 V; and (**e**) −100 V.

**Figure 4 micromachines-14-01800-f004:**
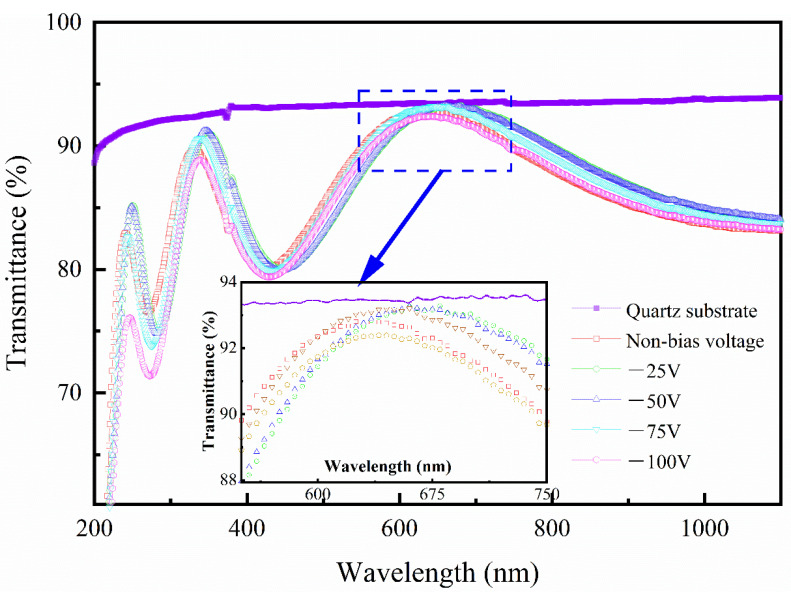
Transmittance spectra for quartz substrate and HfO_2_ thin film deposited by DC reactive magnetron sputtering at different negative bias voltages. The transmittance spectra reveal a larger scale in the wavelength ranging between 550 and 750 nm (insert, bottom center).

**Figure 5 micromachines-14-01800-f005:**
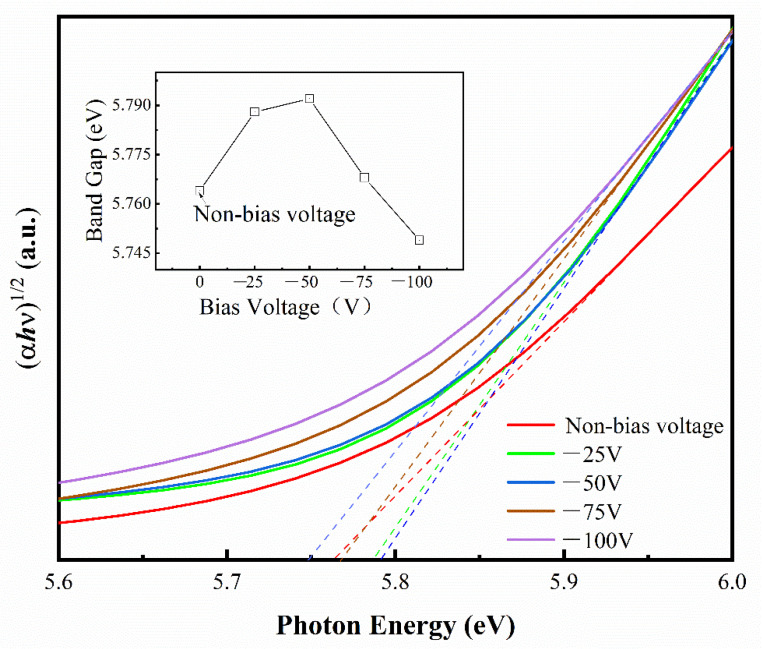
Tauc plot for HfO_2_ thin film deposited by DC reactive magnetron sputtering at different negative bias voltages. Variation in band gap as a function of negative bias voltage (insert, upper-left).

**Figure 6 micromachines-14-01800-f006:**
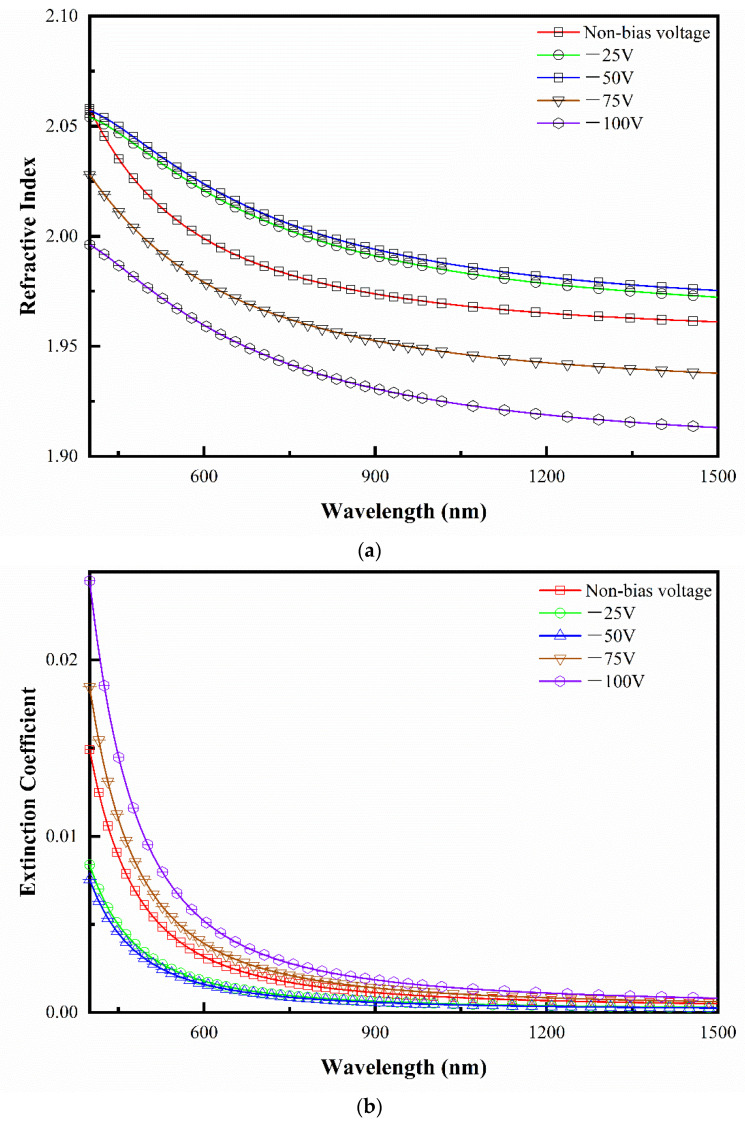
Refractive index (**a**) and extinction coefficient (**b**) for HfO_2_ thin films deposited by DC reactive magnetron sputtering at different negative bias voltages using the Cauchy dispersive model to fit measurement data by spectroscopic ellipsometer in spectral regions ranging between 400 and 1500 nm.

**Figure 7 micromachines-14-01800-f007:**
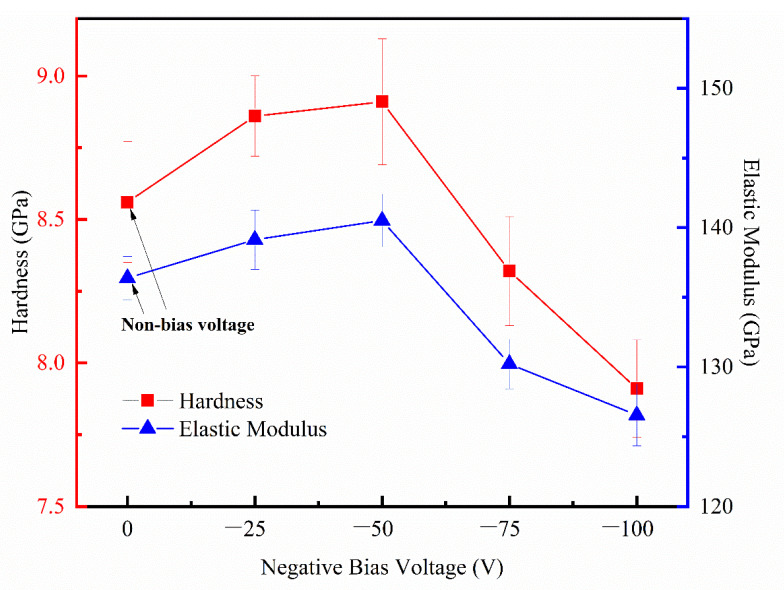
Nano-hardness and elastic modulus of HfO_2_ thin film deposited at different substrate bias.

**Figure 8 micromachines-14-01800-f008:**
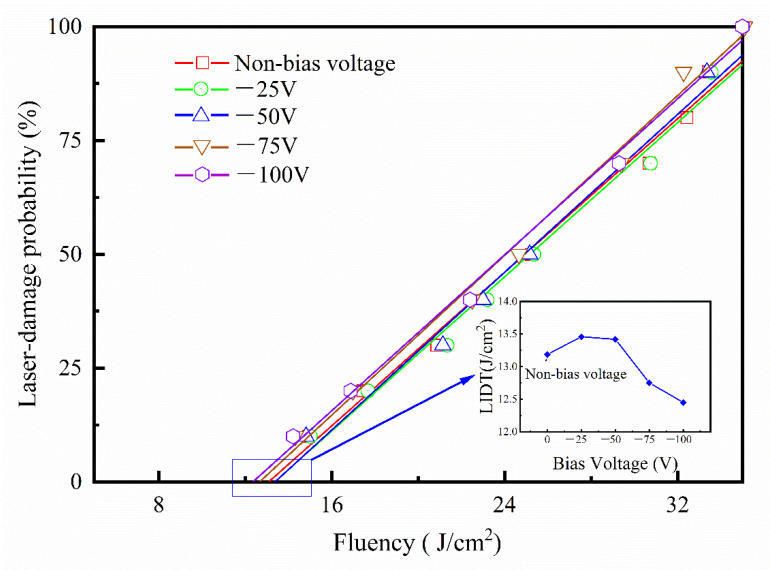
The LIDT of HfO_2_ thin film deposited at different substrate bias.

## Data Availability

The data that support the findings of this study are available from the corresponding author Yingxue Xi upon reasonable request.

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
