# Peer review of "Effect of Substrate Negative Bias on the Microstructural, Optical, Mechanical, and Laser Damage Resistance Properties of HfO2 Thin Films Grown by DC Reactive Magnetron Sputtering"

_micromachines, 2023, doi:10.3390/mi14091800_

Round 1
Reviewer 1 Report (Previous Reviewer 1)
The authors addressed the issues well, now the paper can be accepted for publication.
Author Response
In the revised edition, we have polished the English language as carefully as possible. We would be grateful if there are suggestions for further improvement.
Reviewer 2 Report (New Reviewer)
The content of the article is ok, but there are some formatting errors, grammar errors, inconsistent words, and inconsistent reference formats in the article. After careful revision, publication can be acceptable.
The change in elastic modulus in Figure 7 is significant, which may be due to the testing error. In addition, the explanation of the reasons for the change in elastic modulus is not comprehensive enough and rigorous enough!
The sentence expression in the article needs to be carefully revised!
Author Response
Pls, check the attached document.

Reviewer 3 Report (New Reviewer)
The authors report on a series of experiments aimed at exploring the influence of substrate bias on a variety of physical characteristics of HfO2 thin films, deposited by magnetron sputtering. The topic is of interest both, to the semiconductor device community and to materials scientists. As such, there is merit to this work.
Unfortunately, there are several major deficiencies in the paper, aside from the problems with the English syntax and grammar listed below.
1. The key argument appears to be that the negative bias affects the porosity and thus the density of the films, which then has consequences for the optical and mechanical properties. While this may be a valid argument, the authors do not present even estimates for the film density.
2. Many of the observed changes in physical properties with bias are very small and without experimental uncertainties given, it is impossible to know if the changes are statistically significant. For example, on p. 9, the authors call refractive index changes from 2.01 to 2.03 to 2.03 to 1.99 and to 1.97 "dramatic". What is the accuracy of these values? Are the differences significant? How is any of this "dramatic"?
3. On p. 5, the authors discuss grain sizes "decreasing slightly from approximately 84nm to 81nm, suggesting a grain refinement. As the substrate bias voltage was increased to -75V the grains size continued to decrease to 79nm, and then dropped to 76nm as the negative bias voltage rises -100V". Again, what is the accuracy of these values?
4. On p. 4, with reference to Fig. 2, the authors state "as the negative bias increases from 0 to -100V, the intensity of the dominant peak (−111) decreases....". This is not at all obvious from the figure. In fact, the main peak appears to be largest at -100V. The issue is important as the authors use this observation to infer "...a decline in the average crystallite size and hence crystallinity of the films".
5. On p. 2, the authors state that "...thicknesses of the samples deposited were measured using a surface profilometer from ZYGO...". What was the measurement set-up? Did they measure a step height? What is the accuracy of such measurements?
6. P. 6 with reference to Fig. 4: The authors state that "...it can be noticed that the transmission was decreasing as the negative bias was increased from -50 to -100V...". Again, the effect is very small. How accurate are these measurements? Do the spectral oscillations match with the measured film thickness?
7. On p. 8, with reference to Fig. 5, the authors state "...it can be clearly seen that the optical bandgap ranges from 5.75 nm to 5.79 nm" and call this a "dramatic increase in optical band gap energy along with the increasing of the negative biases". Again, the effect is very small. How accurate are these measurements? How can a change by less than 1% over the entire range be "dramatic"?
The paper is reasonably well structured but very difficult to read. There are many incomplete sentences while at the same time, some sentences run on for 5 or 6 lines. The manuscript also contains many repetitive statements. On several occasions, the authors chose to use very peculiar words, such as, for example, "this paper lucubrates" and "experiments have unfolded". In a revised version the syntax and grammar need to be significantly improved to make the paper better readable.
Round 2
Reviewer 3 Report (New Reviewer)
The authors have tried to address all of my concerns, in many cases, successfully. I still am not convinced that the accuracy of the data allows the complex interpretations offered in the manuscript, especially as the authors in their reply still did not provide sufficient quantitative information on experimental uncertainties.
The quality of the written text has improved.
Author Response
pls check the attachment.

This manuscript is a resubmission of an earlier submission. The following is a list of the peer review reports and author responses from that submission.
Round 1
Reviewer 1 Report
Manuscript ID: micromachines-2509437
The manuscript entitled: " Effect of Bias Voltage on Substrate for the Structure, optical properties, mechanical properties, and laser-induced damage Threshold HfO2 Thin Films grown by Reactive Magnetron Sputtering" by Yingxue Xi et al. is referred to applying different substrate biases during magnetron sputtering deposition process to controlling the crystallization of HfO2 thin films and improving diverse properties of HfO2 thin films via changing the voltages of substrate bias. This is a detailed research work; however, there still have a few flaws in this paper. The authors need to clarify the remarks which are listed below:
Q1:In this paper, the refractive index and extinction coefficient of thin films are measured indirectly by ellipsometry, and the analysis of ellipsometry depends heavily on the model selection, what can be done to determine the correctness of ellipsometric data
Q2: In Figure 3, The UV-Vis-IR spectra of hafnium oxide films deposited at various bias voltages. The authors state that the bias voltage induces a small shift in the peak of the transmission spectrum, implying that the deposition rate of the film is only slightly influenced by the negative bias of the substrate. I think this conclusion is unreasonable, or at least inaccurate, and that transmission spectroscopy can solve for the optical thickness of the film, not the actual physical thickness
Q3: The author's explanation of the variation pattern of laser damage threshold with negative bias voltage points out the correlation between laser threshold magnitude and film refractive index, but does not clarify the reason for the correlation between film refractive index and laser threshold; the paper points out that there is literature reporting the correlation, but the source of the literature is not indicated, please annotate it
Minor editing of English language required
Reviewer 2 Report
The title indicates, that optical properties are investigated as a relevant part in the paper. However, there are some points which are critical and also contain some important mistakes. Comments as follows:
1. Citation at the beginning is focussed on ferroelectrical properties, other citations e.g. on optical properties are missing
2. The author states that ALD has bad properties compared with magnetron sputtering. This must be argued better and not by chosing one paper. It is known that low-loss optical coatings using ALD are ailable today (I do not promote ALD, but one should be more neutral)
3. What is the model used for the roughness layer? Typically a mixture of AIR and the cauchy-layer with some specific amount of AIR is used (with Maxwell-Garnet theory, or Bruggeman effective medium). In the following, the surface roughness is not used any more? Does it play a role?
4. The authors used only oxygen for sputtering the HfOx layers. It is known that by using oxygen only, a crystalline layer growth results. This is by far not the way how optical layers should be deposited, because it will lead to scattering.
5. From figure 3, strong optical losses can be seen. This can either be due to absorption, or because of scattering. I doubt it is only absorption as analysed further in the paper.
6. Resulting from 5., the figure 4 contains wrong results because scattering is not taken into account! In agreement, the absorption coefficient is much too high for typical HfO2 films. Typical values are less than 10-3 or 10-4 for wavelengths larger than 400nm.
7. Nano hardness was measured on films with about 180nm. Therefore, substrate effects may arise. Is this checked?
Reviewer 3 Report
The authors discussed the effect of negative bias to the HfS2 thin film specifically on its microstructure variation upon the negative bias. However, besides discussing the research that they made, the manuscript has serious flaws to be published as an article of Micromachines.
1. The writing of this manuscript must be improved. For example, in the beginning of the introduction the authors wrote "Hafnium oxide (Hafnium dioxide, HfO2), Which can be used as a high refractive index 28 material with a wide transparency from UV, visible to mid-infrared (MIR)." Where is the preposition in the sentence? Such issues are found everywhere in the manuscript.
2. Poor typo is another issue. For example, the author wrote "... high-k dielectric film with a wide band gap (Eg>5.0eV),..." Basically, "Eg>5.0eV" should be written Eg > 5.0 eV. This is a ground rule in writing the manuscript. Such issues, again, are found everywhere in the manuscript.
3. What is the "electron-beam thermal evaporation"? As far as I know, and after checking the reference that the authors cited, that should be one of electron-beam evaporation or thermal evaporation. The authors, again, need to check the manuscript carefully prior to receiving any reviews. Without any checkup, this is simply a waste of time to read this manuscript regardless of what the authors claim.
The manuscript needs extensive corrections not only on the grammatical issues but only on the ground rule in preparing the manuscript.